# DTBV: A Deep Transfer-Based Bone Cancer Diagnosis System Using VGG16 Feature Extraction

**DOI:** 10.3390/diagnostics13040757

**Published:** 2023-02-16

**Authors:** G. Suganeshwari, R. Balakumar, Kalimuthu Karuppanan, Sahaya Beni Prathiba, Sudha Anbalagan, Gunasekaran Raja

**Affiliations:** 1School of Computer Science and Engineering, Chennai Campus, Vellore Institute of Technology, Chennai 600127, India; 2NGNLabs, Department of Computer Technology, MIT Campus, Anna University, Chennai 600044, India; 3Department of Biotechnology, School of Bioengineering, Kattankulathur Campus, SRM Institute of Science and Technology, Chennai 603203, India; 4Center for Smart Grid Technologies, School of Computer Science and Engineering, Vellore Institute of Technology, Chennai 600127, India

**Keywords:** health care, bone cancer, X-ray image, medical image processing, convolutional neural network, transfer learning, support vector machines

## Abstract

Among the many different types of cancer, bone cancer is the most lethal and least prevalent. More cases are reported each year. Early diagnosis of bone cancer is crucial since it helps limit the spread of malignant cells and reduce mortality. The manual method of detection of bone cancer is cumbersome and requires specialized knowledge. A deep transfer-based bone cancer diagnosis (DTBV) system using VGG16 feature extraction is proposed to address these issues. The proposed DTBV system uses a transfer learning (TL) approach in which a pre-trained convolutional neural network (CNN) model is used to extract features from the pre-processed input image and a support vector machine (SVM) model is used to train using these features to distinguish between cancerous and healthy bone. The CNN is applied to the image datasets as it provides better image recognition with high accuracy when the layers in neural network feature extraction increase. In the proposed DTBV system, the VGG16 model extracts the features from the input X-ray image. A mutual information statistic that measures the dependency between the different features is then used to select the best features. This is the first time this method has been used for detecting bone cancer. Once selected features are selected, they are fed into the SVM classifier. The SVM model classifies the given testing dataset into malignant and benign categories. A comprehensive performance evaluation has demonstrated that the proposed DTBV system is highly efficient in detecting bone cancer, with an accuracy of 93.9%, which is more accurate than other existing systems.

## 1. Introduction

There are over 100 hufman metabolic diseases worldwide [1]. Among these diseases, cancer is a significant threat to humanity due to its lethal nature. Cancer is indicated by abnormal cell growth and rapidly spreads to other body parts. There are a variety of factors that cause cancer, some of which are preventable and others that are not. For example, factors such as smoking, drinking alcohol, and exposure to excessive ultraviolet (UV) radiation are preventable. On the other hand, genetic factors are not preventable [2]. These genetic factors are either inherited from parents or caused due to changes in their lifestyle, such as the usage of tobacco products. In addition, age is the most prominent among the unpreventable factors.

Bone cancer is a rare medical condition that may begin with swelling or tenderness in the affected area of the bone. Bone cancer occurs when abnormal cells in the bone grow out of control and may spread to other parts of the body. The number of new cases of bone cancer is likely to increase by 3900 in 2022, with 2100 deaths expected as well [3]. Bone cancer occurs most commonly in people below the age of 20. Radiography, X-ray, computed tomography (CT) scans, magnetic resonance imaging (MRI), ultrasound, phototherapy, and positron emission tomography (PET) are some of the common modalities used in medical imaging [4,5]. Our proposed method, A Deep Transfer-Based Bone Cancer Diagnosis System Using VGG16 Feature Extraction (DTBV) system, employs an X-ray imaging modality. Because they are affordable and widely available, medical X-ray images constitute a substantial and valuable resource for research and disease diagnosis [6].

In recent years, various treatments have been proposed to treat this disease, including surgery, radiation therapy, chemotherapy, and targeted therapies. Identification of cancer in the bone is a testing issue owing to its complex structure [7]. The manual examination of cancerous images is cost-ineffective due to costly equipment and sometimes results in erroneous results due to the mishandling of details. Machine learning (ML) is now frequently used in cancer research as it provides an easy way to perform data analysis and automated extraction of important information from the data [8].

Pre-processing medical images is a fundamental technique used to improve the performance of any ML model [9]. Pre-processing helps improve image quality and thus helps us analyze the data more effectively. The common image pre-processing techniques are resizing, filtering, segmentation, pixel brightness transformations, data augmentation, and normalization.

For the development of clinical decision models, support vector machine (SVM) classification approaches have been widely used [10]. An SVM is an ML-based classification technique used to classify X-ray and other modality images with minimal computation. In classification, the SVM model appears more accurate than the convolutional neural network (CNN) model, especially when the datasets are small [11]. CNN is an artificial neural network (ANN) used for automated image recognition and object detection [12]. Our proposed DTBV system employs resizing and filtering for pre-processing and combines the CNN and SVM models.

In the proposed DTBV system, a transfer learning (TL) approach is used in which the features are extracted using the VGG16 model and then selected based on the mutual information statistic. This feature selection method has been used to detect bone cancer for the first time. Features with stronger correlations with the target variable have more predictive power, and such features are selected using this method, reducing the model’s overfitting. The TL approach involves using a trained model for one task to be repurposed to perform another related task. The features are then fed to the SVM classifier that separates the input dataset into healthy and malignant data. The VGG16 model, a type of CNN model with 16 layers in its architecture, is mainly used for its automatic feature extraction capability. It is trained on 14 million images belonging to 22,000 categories from the ImageNet dataset. Instead of using the entire VGG16 model, only a few network layers are used for feature extraction. The weights of the pre-trained layers are kept fixed because the pre-trained layers contain useful features learned from the previous task and can be reused for the new task. The proposed DTBV system is designed to overcome the current limitation of the cumbersome manual technique with improved accuracy in detecting cancerous images.

The key contributions of this paper are summarized as follows:Developing a DTBV system that can assist doctors by providing a second opinion to improve diagnostic efficiency by comparing malignant and healthy bone images in real-time.Prediction of bone cancer at an early stage using the VGG16 model for feature extraction, mutual information statistics for feature selection, and the SVM model for classification with X-ray images as a modality.A comparative study on the performance of various CNN models for feature extraction and ML models for classification, with further consideration to the performance of the proposed DTBV system with other existing systems.

The rest of this article is organized as follows. Section 2 includes a brief study of related studies. Section 3 describes the proposed DTBV system. Section 4 delivers the report of the experimental setup, demonstrates the results, and discusses the performance achieved. Finally, Section 5 concludes this article.

## 2. Materials and Methods

The development of ML models has significantly contributed to cancer prediction and diagnosis [13]. Medical images usually contain a lot of noise, which may affect the post-processing of images. As a result, the denoising of input images appears to be the first and most crucial stage in the pre-processing of input images. The amount of noise should be specific, as removing too much noise may leave important details behind. On the other hand, removing very little noise may result in undesired output. As a solution to this issue, a two-stage deep CNN method for medical image denoising is proposed in [14], wherein the image and noise components of the medical image are taken into account equally, and the image denoising task was formulated as an image decomposition problem. The strategy proposed by the authors of [15] was another approach to noise removal. The strategy utilized a novel min–max average pooling-based filter to remove salt and pepper noise. This approach improved the peak signal-to-noise ratio (PSNR) in denoising medical images corrupted by medium to high noise densities by 1.2 dB. The authors of [16] propose a method for bone cancer detection using simple statistical feature extraction and SVM-based computerized classification. They used the maximum value among the mean and median values to replace each pixel for filtering. Their approach was found to be 92% accurate.

The use of ML goes beyond medical imaging to other fields as well. In this paper [17], the authors developed an unsupervised ML strategy for optimizing the route taken by self-driving vehicles from their starting points to their destination. The strategies include self-organizing mapping, hierarchical gaussian matrix models, and clustering-based K-Means. With the real-time parallel implementation of the unsupervised ML algorithms, the autonomous vehicle was able to respond in under one microsecond to lateral, longitudinal, and angular motion changes; it was also able to contribute to less traffic congestion and minimized collisions. Another engineering application of ML is proposed in [18]. This paper proposes an efficient solution for antenna design automation based on ML-based surrogate-assisted particle swarm optimization (SAPSO). The suggested solution closely combines two ML-based approximation models and particle swarm optimization (PSO). Then, to identify potential candidates for full-wave electromagnetic (EM) simulations, a novel mixed prescreening (mix P) technique is proposed. After verifying with three real-world antennas, the results show that the proposed SAPSO–mix P technique can find favorable results with a much smaller EM simulation than other existing methods.

Feature extraction is essential in image processing, data mining, and computer vision applications [19]. The authors of [20] propose a system for identifying bone cancer based on a Gray Level Co-occurrence Matrix-based (GLCM) textural feature. They used two ML models, random forest and SVM, to run two trials, one using hog feature sets and the other without them. The trial that used the SVM model with hog feature sets reported an accuracy of 92.5%. However, using a CNN seems to be a better way of extracting features [21]. The authors of [22] propose an approach of CNN-based feature extraction for Coronavirus Disease of 2019 (COVID−19) detection. They developed a deep uncertainty-aware transfer learning framework in which four popular CNNs were applied to extract features from chest X-ray images. Different ML models were then used to classify the extracted features. A comparison of various simulations on X-ray image datasets indicated that VGG16 and ResNet50 as feature extractors and linear SVM and multilayer perceptron as classifiers outperformed other neural network models.

Classification is the final process in any ML model or system. The authors of [23] propose an approach for detecting bone cancer using fuzzy C-mean clustering and Adaptive. The Neuro Fuzzy Inference System (ANFIS) is used to classify benign and malignant bone cancer. They evaluated their model using three performance measures, namely accuracy, sensitivity, and specificity, and the corresponding measures were found to be 93.7%, 87.5%, and 100%, respectively. The authors of [9] propose another approach for classifying bone tumors in the proximal femur area using the EfficientBet-b2 CNN model. They have used min–max normalization and data augmentation for pre-processing. They evaluated their model using five-fold cross-validation and obtained an accuracy of 85.3%, a sensitivity of 82.2%, and a specificity of 91.2%. The authors of [24] emphasized the need to develop a DNN model to classify fractured and healthy bones. To increase the size of the dataset in this work and prevent overfitting, data augmentation techniques were applied. This proposed DNN model, which used softmax and the Adam optimizer, shows an accuracy of 92.4% when evaluated using five-fold cross-validation. The authors of [25] propose an efficient approach that quickly classifies bone scintigraphy images of prostate cancer patients by determining whether they develop prostate cancer metastasis. The proposed method classifies the data into three categories: malignant, healthy, and degenerative. After various exploration and experiments, the proposed CNN architecture consists of 4 convolutional-pooling layers, two dense layers followed by a dropout layer, and a final output layer with three nodes. In the initial and the intermediate layers, ReLU activation is used, followed by the Adam optimizer at the output nodes. The results showed that the method is sufficiently accurate, with an accuracy of 91.61% when distinguishing a bone metastasis instance from other cases of degenerative alterations or normal tissue.

In addition to computer science, CNN is utilized in many other fields. The authors of this paper [26] propose a customized CNN with various hyperparameter tuning for crack detection in concrete structures. The proposed method enables automatic crack detection, which is very useful when inspecting concrete structures. They compared their own customized CNN with existing pre-trained models and achieved better results in terms of accuracy and precision. An efficient approach to detect anomalies in Industrial Control Systems (ICSs) traffic is proposed in [27]. This paper presents a model based on a deep residual CNN to prevent gradient explosion or gradient disappearance. A modified residual CNN architecture combined with the TL approach ensures that unknown attacks can be detected. The model gives reliable predictions for unknown and abnormal data through short-term training. The proposed method gives a higher score and solves the time problem associated with model training compared to existing methods.

Improving feature extraction efficiency and coupling it with higher classification accuracy is critical in constructing an effective bone cancer detection model that improves the diagnosis technique. The proposed DTBV system aims to combine the advantages of the VGG16 model in extracting useful features from X-ray images with SVM in image classification and improve the performance of detecting bone cancer more effectively compared to existing systems. In this system, features are selected using mutual information statistics, a technique that has not been utilized for feature selection in existing bone cancer detection systems.

## 3. Proposed System

This paper proposes a DTBV system based on the hybrid CNN–SVM architecture for detecting bone cancer using the VGG16 model, a standard deep CNN architecture with multiple layers, as a feature extractor and SVM to train over these features. The proposed DTBV system can detect cancerous images on time while improving accuracy. The overview of the proposed DTBV system is shown in Figure 1.

The majority of image datasets contain noise. As a result, they are pre-processed with a median filter to reduce the noise of extreme magnitudes. This noise reduction is crucial since it aids in the improvement of subsequent processing outcomes. The median filter sharpens the image dataset when given as input while preserving the edge.
*I*(*x*, *y*) ← *median*{*N*(*I*)}(1)
where *N*(*I*) are the neighboring pixels of the image.

The VGG16 model extracts features from the filtered images through which relevant data are obtained. The VGG16 model is primarily used for image processing and classification tasks. The VGG16 model generates features automatically, which are then integrated with the classifier. In the proposed DTBV system, for feature extraction, only convolutional layers, pooling layers, and the first fully connected layers are used, while the convolution and max-pooling layers are arranged consistently [28]. The convolutional layer is the first layer that extracts features from the image data. This layer performs the convolution operation on the input image.
(*I* ⊕ *K*) [*p*, *q*] ← Σ*_m_*Σ*_n_ I* [*p* − *m*, *q* − *n*] *K*[*m*, *n*](2)
where ⊕ is the convolution operation, *K* is the kernel matrix, and (*p*, *q*) is the dimension of the resultant feature map. A convolution operation combines all the pixels in the receptive field into a single value. This layer is coupled with the ReLU activation function. With this activation function, the VGG16 model improves its learning speed.
*I* ⊕ *K* ← *max*(0, *I* ⊕ *K*)(3)
f ← *I* ⊕ *K*(4)

The feature map is obtained as the output from the convolutional layer, which is then passed as input to the max-pooling layer. The max-pooling layer performs dimensionality reduction, wherein the number of parameters in the feature map from the convolutional layer is reduced according to Equation (5).
(5)$$\left(W_{o},H_{o},D_{o}\right)\leftarrow \left(\frac{W_{i}-k}{S}+1,\frac{H_{i}-k}{S}+1,D_{i}\right)$$
where (*W_o_*, *H_o_*, *D_o_*) are the output dimensions of the feature map, (*W_i_*, *H_i_*, *D_i_*) are the input dimensions of the feature map, *k* is the kernel size, and *S* is the stride value.

This layer further helps to reduce complexity and limits the risk of overfitting. Fully connected layers are the final layers of the network, and the output from the final pooling or convolutional layer is flattened and subsequently fed into the fully connected layer.

The best features are then selected from the feature map, obtained from the first fully connected layer, using mutual information statistics that measure how much one random feature tells us about another.
*fs* ← *feature_selector* (*f*)(6)

The information gain, measured by the entropy, between various features is used to calculate the mutual information statistics. Feature selection is crucial as it eliminates redundant predictors from the model, further improving accuracy and reducing training time.

In the proposed DTBV system, SVM, a supervised learning algorithm, is used for classification.
*C* ← *classifier* (*fs*)(7)

The selected features are split into training and testing sets. The training set is fed into the SVM model and the testing set is used to classify the images as healthy and cancerous bones.

The overall summary of the proposed DTBV system is shown in Algorithm 1.
**Algorithm 1:** Feature Extraction and Classification Using the DTBV System**Input:** Bone X-ray image I
**Output:** Classified image *C*
1. **procedure** FEATURE_EXTRACTION(*dataset*)
2.     **for**
*I* in *dataset*
**do**
3.          Read the image 
              *I*←*cv2.imread*(*image_path*)
4.          Resize the image
5.          Apply median filter to remove noise from the image 
              *I*←*cv2.medianBlur*(*I*, *3*)
6.          Extract the features from the filtered image using the VGG16 model 
              *feature_extractor*←*vgg16*() 
              *f*←*feature_extractor*(*I*)
7.     **end for**
8. **end procedure**
9. **procedure** CLASSIFICATION(*dataset*, *f*)
10.     Select the best features from the extracted features using mutual information statistic 
            *f s*←*SelectKBest*(*mutual_info_classif*)
11.     *Split* the dataset into training_dataset and testing_dataset
12.     Train the SVM model with the features selected for the *training*_*dataset*

              *classifier*←*SVC*()
              C←*classifier*(*fs*)
13.     Classify the *testing*_*dataset* using the selected features into healthy and malignant images
14. **end procedure**

## 4. Results and Discussions

### 4.1. Experimental Setup

The proposed DTBV system is implemented using bone cancer. X-ray image dataset acquired from the repository of the Indian Institute of Engineering Science and Technology, Shibpur (IIEST) [20], on Google Colab with approximately 13 GB of RAM and 110 GB of disk space. The Google Colab environment is further equipped with a high-performance NVIDIA Tesla K80 GPU. The input dataset contains 100 images, 50 of which are healthy bone images and 50 of which are malignant bone images. The input images are resized to 255 × 255 pixels to establish the base size and uniformity [29]. Figure 2 and Figure 3 show the sample images of healthy and malignant bones. The OpenCV library is used to implement the median filter. The PyTorch library is used to implement the VGG16 model for feature extraction. The sklearn library is used to perform feature selection and to implement the SVM model for classification.

### 4.2. Results for Pre-Processing

In the initial stage, the input image, as shown in Figure 4a, is resized. Then, median filter is applied to the resized image as seen in Figure 4b. As a result, we get a sharper image, as shown in Figure 4c, with reduced noise.

### 4.3. Results for Feature Extraction and Classification

A feature extraction process transforms the input data into numerical features while maintaining the information given in the original data. The filtered images are run into the VGG16 model at this stage to extract the features. As a result, 4096 features are extracted. With a 2:1 split ratio, the dataset is divided into 67 training images and 33 testing images.

The SVM model is trained with the training dataset, and the testing dataset is classified as healthy or malignant bone. As a result, the confusion matrix of test data is shown in Figure 5. The confusion matrix summarizes the prediction results performed by the proposed DTBV system on testing data. Out of 18 healthy bones, 1 is incorrectly identified as cancerous, and 17 are correctly identified as healthy. Out of 15 cancerous bones, 1 is incorrectly identified as healthy, and 14 are correctly identified as cancerous.

### 4.4. Comparison of Model Performance

A crucial component of the ML process is performance evaluation. After classification, the performance is evaluated with the following indicators: accuracy, recall, false-positive rate (FPR), specificity, precision, and F1-score [30]. The accuracy of the proposed DTBV system is compared with various CNN models for feature extraction, namely VGG19, InceptionV3, and ResNet50, as shown in Table 1.

From the results obtained, we infer that the VGG16 model employed in the proposed DTBV system outperforms other CNN models in extracting the features.

The performance of the proposed DTBV system is compared with four other ML models, namely logistic regression, random forest, k-nearest neighbor (KNN), and decision tree for classification, as shown in Figure 6a.

Overall, the proposed DTBV system, which employs SVM for classification, performs better than other models. The assessment of the performance metrics of the proposed DTBV system with other ML models for classification is in Table 2.

The receiver operating characteristic (ROC) curves for the proposed DTBV system, along with four other ML models, are shown in Figure 6b. Better performance is obtained as the curve moves closer to the top-left corner. The curve obtained for the proposed DTBV system is much closer to the top-left corner than the curves of the other models, indicating that it is significantly better.

The area under the curve (AUC) measures the model’s ability to distinguish between healthy and cancerous images accurately. The ability to distinguish between healthy and cancerous images is more for higher AUC. The AUC for the proposed DTBV system is evaluated to be 93.9%, whereas logistic regression, random forest, KNN, and decision tree have yielded 88.3%, 76.7%, 82.2%, and 86.1%, respectively.

The precision–recall curves for the proposed DTBV system, along with four other ML models, are shown in Figure 6c. Precision and recall are generally better when the area under the curve is a higher value. As we can see, the area under the curve for the proposed DTBV system is much larger than the other models, indicating that it is significantly better.

Finally, the performance of the proposed DTBV system is compared with other existing systems and found to be more accurate in bone cancer classification, as shown in Table 3. Figure 7 compares the proposed DTBV system with other existing systems.

## 5. Conclusions and Future Works

Bone cancer is a rare medical disorder that could spread to other body parts. Manual bone cancer screening is challenging and requires a lot of specialized knowledge to provide sufficient accuracy and reliability. A DTBV system based on a hybrid CNN–SVM model was proposed to address this issue using X-ray images. The proposed DTBV system utilizes mutual information statistics for feature selection, a technique not currently used for bone cancer detection. By comparing the various models, the VGG16 model was found to be the best fit for feature extraction and SVM for classification among the other models compared. The proposed system yielded an accuracy level of 93.9%, which is higher when compared to other existing systems.

The feasibility of the model can be appropriately ascertained in the future in the following directions: (i) The prediction rate of the proposed DTBV system could be increased by training it on larger datasets instead of smaller ones. (ii) The various imaging modalities can be considered to develop an enhanced diagnostic system for various modalities. This will show the proposed method’s ability to be generalized.

(iii) We can also integrate the DTBV system into any sensor-based internet of things (IoT) device for the unique possibility of remote patient monitoring and the enablement of blockchain technology renowned for its decentralized and secure nature. Other comparable technologies can be considered in conjunction with those above, paying special attention to security aspects. (iv) In typical bone cancers such as osteosarcoma, patients are likely to have parts of their bone replaced with bone grafting. The proposed DTBV system could be further improved through the identification of regions of the bone that are malignant as opposed to finding the entire bone to be malignant. Such regions may be identified using segmentation, and a 3D printer could be used to print the malignant regions to replace the malignant bones.

## Figures and Tables

**Figure 1 diagnostics-13-00757-f001:**
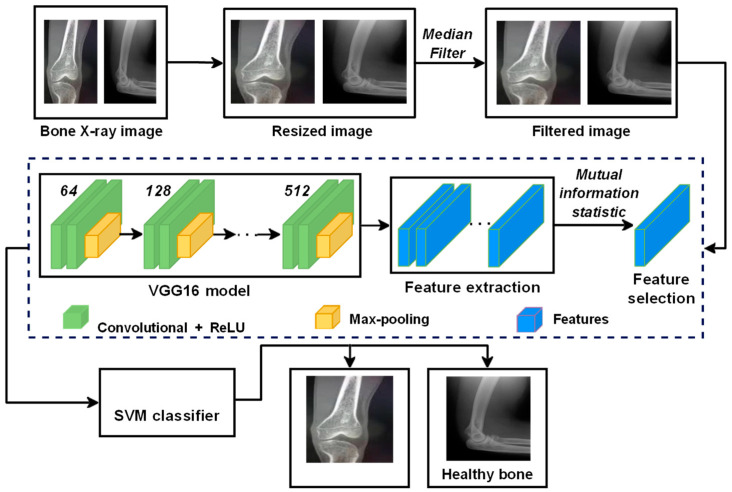
Overview of the proposed DTBV system for the detection of bone cancer.

**Figure 2 diagnostics-13-00757-f002:**
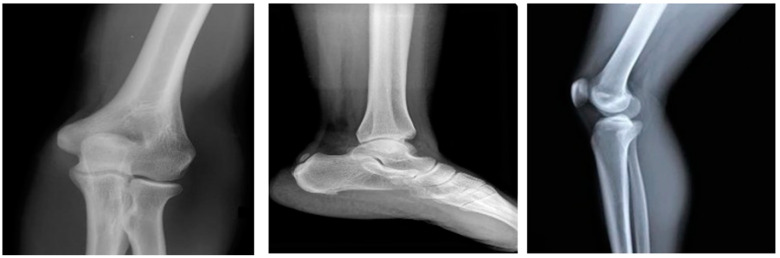
Healthy bones.

**Figure 3 diagnostics-13-00757-f003:**
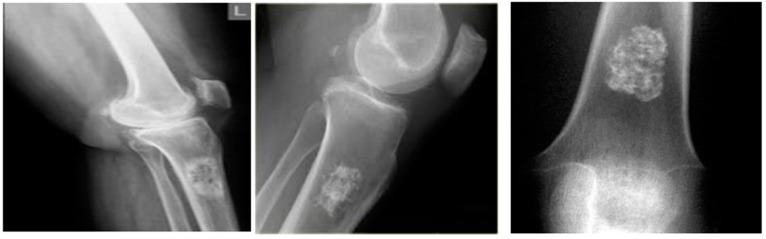
Malignant bones.

**Figure 4 diagnostics-13-00757-f004:**
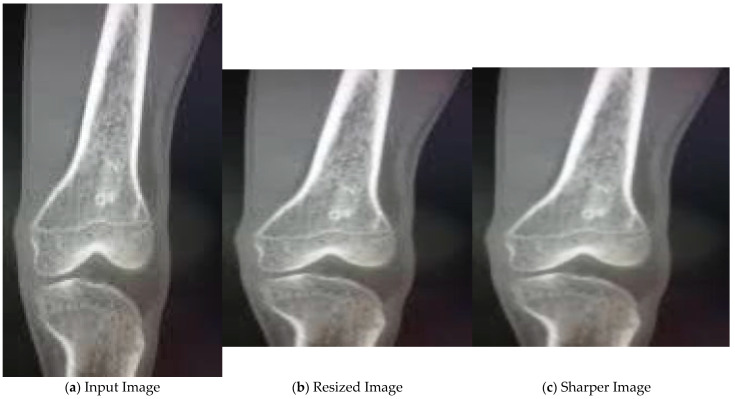
Pre-processing of X-ray images.

**Figure 5 diagnostics-13-00757-f005:**
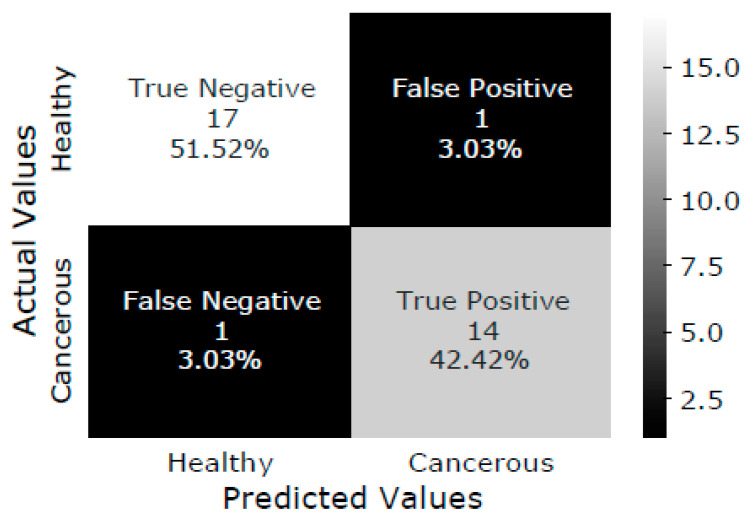
Confusion matrix of test data.

**Figure 6 diagnostics-13-00757-f006:**
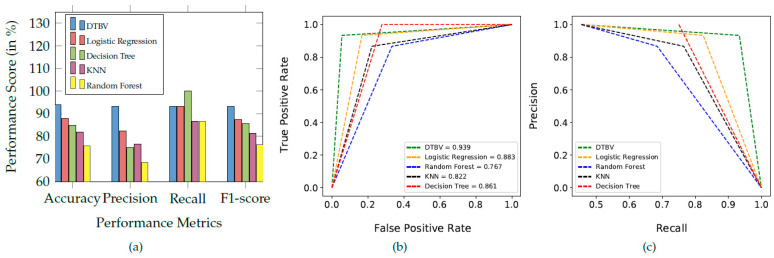
(**a**) Comparison of the proposed DTBV system with other ML models for classification. (**b**) The ROC curves of different classification models. (**c**) Precision–recall curve obtained for the different classification models.

**Figure 7 diagnostics-13-00757-f007:**
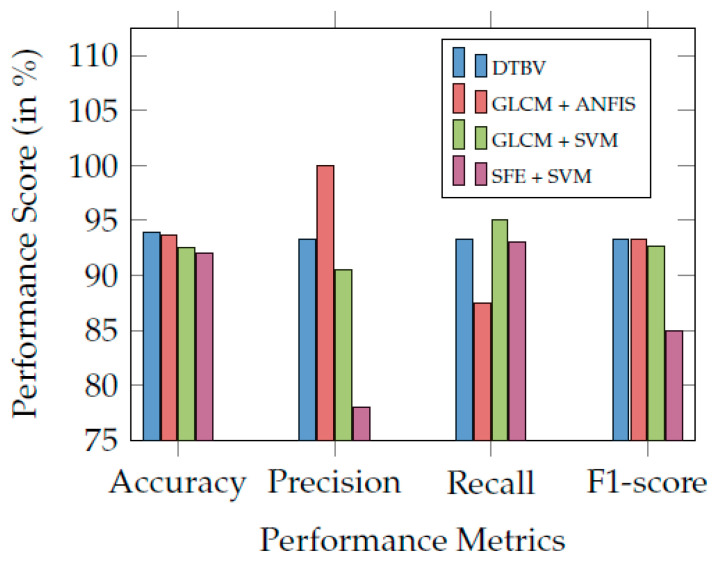
Comparison of the proposed DTBV system with other existing systems.

**Table 1 diagnostics-13-00757-t001:** Comparison of the Proposed DTBV System with Various CNN Models for Feature Extraction.

Model	Accuracy (%)
DTBV	93.9
InceptionV3	81.8
VGG19	79.2
ResNet50	76.2

**Table 2 diagnostics-13-00757-t002:** Comparison of the proposed DTBV system with various ML models for classification.

Performance Metric (%)	DTBV	Logistic Regression	Decision Tree	KNN	Random Forest
Accuracy	93.9	87.9	84.8	81.8	75.8
Recall	93.3	93.3	100	86.7	86.7
Specificity	94.4	83.3	72.2	77.8	66.7
Precision	93.3	82.4	75	76.5	68.4
FPR	5.6	16.7	27.8	22.2	33.3

**Table 3 diagnostics-13-00757-t003:** Comparison of the proposed DTBV system with various ML models for classification.

Performance	DTBV	GLCM +	GLCM +	SFE + SVM [16]
Metric (%)		ANFIS [23]	SVM [20]	
Accuracy	93.9	93.7	92.5	92.0
Recall	93.3	87.5	95.0	93.0
Precision	93.3	100.0	90.5	78.0
F1-score	93.3	93.3	92.7	85.0

## Data Availability

X-ray image dataset acquired from the repository of the Indian Institute of Engineering Science and Technology, Shibpur (IIEST) [20], on Google Colab with approximately 13 GB of RAM and 110 GB of disk space.

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
