# Peer review of "DTBV: A Deep Transfer-Based Bone Cancer Diagnosis System Using VGG16 Feature Extraction"

_diagnostics, 2023, doi:10.3390/diagnostics13040757_

Round 1

Reviewer 1 Report

This manuscript proposed a novel deep learning-based method for bone cancer diagnosis. In the proposed method, VGG16 was established for feature extraction using transfer learning technique, and a SVM classifier was developed for classifying cencerous and healthy bone. The experimental dataset was used to train and validate the performance of the proposed method, with satisfactory results. Overall, the topic of this study is interesting, and the manuscript was well organised and written. The detailed comments are provided below.

1.       The major contributions and novelty of this study should be clearly highlighted in abstract and introduction.

2.       Broaden and update literature review on CNN or machine learning in engineering applications, such as image processing and data analysis. E.g. Torsional capacity evaluation of RC beams using an improved bird swarm algorithm optimised 2D convolutional neural network; Vision-based concrete crack detection using a hybrid framework considering noise effect.

3.       In this study, VGG16 was used for feature extraction. However, it was developed for other task. Hence, transfer learning was employed. Please give more details on how VGG16 was transferred to the model that was used for cancer bone classification.

4.       There are several parameters for both VGG16 and SVM. How did the authors set the optimal values to achieve the best performance of trained model.

5.       There are some mistakes in Figure 6 (b) and (c). Please revise them.

6.       The authors only used 67 image data to trained model. However, the network for feature extraction is VGG16, with deeper architecture. I think the trained network may be overfitted. Please clarify this problem.

7.       How about the robustness of the proposed method against noise effect?

8.       More future research should be included in conclusion part.

Author Response

Point 1: The major contributions and novelty of this study should be clearly highlighted in abstract and introduction.

Response 1: We thank the reviewer for their suggestions. We have highlighted the significant contributions in abstraction and introduction.

Updated Paragraph:

Abstract: Among the many different types of cancer, bone cancer is the most lethal and least prevalent. More cases are reported each year. Early diagnosis of bone cancer is crucial since it helps limit the spread of malignant cells and reduce mortality. The manual method of detection of bone cancer is cumbersome and requires specialized knowledge. A DTBV: Deep Transfer-based Bone cancer diagnosis system using VGG16 Feature Extraction is proposed to address these issues. The proposed DTBV system uses a Transfer Learning (TL) approach in which a pre-trained Convolutional Neural Network (CNN) model is used to extract features from the pre-processed input image, and a Support Vector Machine (SVM) model is used to train these features to distinguish between cancerous and healthy bone. CNN is applied to the image datasets as it provides better image recognition with high accuracy when the layers in neural network feature extraction increase. In the proposed DTBV system, the VGG16 model extracts the features from the input X-ray image. A mutual information statistic that measures the dependency between the different features is then used to select the best features. It is the first time this method has been used for detecting bone cancer. Once features are selected, they are fed into the SVM classifier. The SVM model classifies the given testing dataset into malignant and benign categories. A comprehensive performance evaluation has demonstrated that the proposed DTBV system is highly efficient in detecting bone cancer, with an accuracy of 93.9%, which is found to be more accurate compared to other existing systems.

Introduction: In the proposed DTBV system, a Transfer Learning (TL) approach is used in which the features are extracted using the VGG16 model and then selected based on the mutual information statistic. This feature selection method has been used to detect bone cancer for the first time. Features having stronger correlations with the target variable have more predictive power, and such features are selected using this method and hence reduce the overfitting of the model. The TL approach involves using a trained model for one task to be repurposed to perform another related task. The features are then fed to the SVM classifier that separates the input dataset into healthy and malignant ones. The VGG16, a type of CNN model with 16 layers in its architecture, is mainly used for its automatic feature extraction capability. It is trained on 14 million images belonging to 22,000 categories from the ImageNet dataset. Instead of using the entire VGG16 model, only a few network layers are used for feature extraction. The weights of the pre-trained layers are kept fixed. This is done because the pre-trained layers contain useful features learned from the previous task and can be reused for the new task. The proposed DTBV system is designed to overcome the current limitation of the cumbersome manual technique with improved accuracy in detecting cancerous images.

Point 2: Broaden and update literature review on CNN or machine learning in engineering applications, such as image processing and data analysis. E.g. Torsional capacity evaluation of RC beams using an improved bird swarm algorithm optimised 2D convolutional neural network; Vision-based concrete crack detection using a hybrid framework considering noise effect.

Response 2: We thank the reviewer for their suggestions. We have updated the literature review.

Added Literature Review for the following papers:

  1. W. Abegaz, “A Parallelized Self-Driving Vehicle Controller Using Unsupervised Machine Learning,” in IEEE Transactions on Industry Applications, vol. 58, no. 4, pp. 5148-5156, July-Aug. 2022.
  2. Ali, F. Alnajjar, H. A. Jassmi, M. Gocho, W. Khan, and M. A. Serhani, “Performance Evaluation of Deep CNN-Based Crack Detection and Localization Techniques for Concrete Structures,” in Sensors, vol. 21, no. 5, p. 1688, Mar. 01, 2021.
  3. Fu, X. Cai, B. Yuan, Y. Yang and X. Yao, "An Efficient Surrogate Assisted Particle Swarm Optimization for Antenna Synthesis," in IEEE Transactions on Antennas and Propagation, vol. 70, no. 7, pp. 4977-4984, July 2022, doi: 10.1109/TAP.2022.3153080.
  4. Wang, Z. Wang, Z. Zhou, H. Deng, W. Zhao, C. Wang, and Y. Guo, “Anomaly detection of industrial control systems based on transfer learning,” in Tsinghua Science and Technology, vol. 26, no. 6, pp. 821–832, December 2021.
  5. Papandrianos, E. Papageorgiou, A. Anagnostis, and K. Papageorgiou, “Efficient Bone Metastasis Diagnosis in Bone Scintigraphy Using a Fast Convolutional Neural Network Architecture,” Diagnostics, vol. 10, no. 8. MDPI AG, p. 532, 30-Jul-2020.

Point 3: In this study, VGG16 was used for feature extraction. However, it was developed for other task. Hence, transfer learning was employed. Please give more details on how VGG16 was transferred to the model that was used for cancer bone classification. 

Response 3: We thank the reviewer for their suggestions. We have added how Transfer Learning was employed.

Updated Paragraph:

Introduction: In the proposed DTBV system, a Transfer Learning (TL) approach is used in which the features are extracted using the VGG16 model and then selected based on the mutual information statistic. This feature selection method has been used to detect bone cancer for the first time. Features having stronger correlations with the target variable have more predictive power, and such features are selected using this method and hence reduce the overfitting of the model. The TL approach involves using a trained model for one task to be repurposed to perform another related task. The features are then fed to the SVM classifier that separates the input dataset into healthy and malignant ones. The VGG16 model, a type of CNN model with 16 layers in its architecture, is mainly used for its automatic feature extraction capability. It is trained on 14 million images belonging to 22,000 categories from the ImageNet dataset. Instead of using the entire VGG16 model, only a few network layers are used for feature extraction. The weights of the pre-trained layers are kept fixed. This is done because the pre-trained layers contain useful features learned from the previous task and can be reused for the new task. The proposed DTBV system is designed to overcome the current limitation of the cumbersome manual technique with improved accuracy in detecting cancerous images.

Point 4: There are several parameters for both VGG16 and SVM. How did the authors set the optimal values to achieve the best performance of trained model.

Response 4:  We thank the reviewer for their suggestions. 

For the Feature extraction part, we used only the convolutional layers and the max-pooling layers, along with the first fully-connected layer in the pre-trained VGG16 model. For the classification part, we used a combination of values for random state for SVM, KNN, Decision Tree, Logistic Regression and Linear Regression, and we picked the value of random state for which maximum accuracy is obtained, and the RMSE value is minimized.

Point 5: There are some mistakes in Figure 6 (b) and (c). Please revise them.

Response 5:  We thank the reviewer for their suggestion.

We have checked figures 6 (b) and (c). In figure 6(c), the recall value for the decision tree is 100%, which is why it starts slightly away from the y-axis. And both the figures were plotted with the values we got from the model itself.

Point 6: The authors only used 67 image data to trained model. However, the network for feature extraction is VGG16, with deeper architecture. I think the trained network may be overfitted. Please clarify this problem.

Response 6: We thank the reviewer for their suggestions.

To solve the overfitting issue, we selected the best random state value and utilized the feature selection method later on to prevent overfitting. We used the mutual information statistic-based feature selection method, which is our new approach, as no one has utilized this feature selection method for detecting bone cancer. And we reduced the number of features, and features having stronger correlations with the target variable have more predictive power, and such features are selected using this method. Hence overfitting is resolved.

Point 7: How about the robustness of the proposed method against noise effect?

Response 7:   All X-ray images contain salt and pepper noise. The median filter is applied for removing such noises as it gives the best result compared to other filters. In the median filter, it picks the median value from the neighboring window size, and as a result, salt (low-level values – 0) and pepper noise (high-level values – 1) are entirely removed. We are using the median filter in the proposed system to filter the input x-ray images, and hence noise effect could be removed easily.

Point 8: More future research should be included in conclusion part.

Response 8:  We thank the reviewer for their suggestion. We have added about future research part in the conclusion part.

Updated Paragraph:

In typical bone cancers like osteosarcoma, patients will likely have parts of their bone replaced with bone grafting. The proposed DTBV system could be further improved through the identification of regions of the bone that are malignant as opposed to finding the entire bone to be malignant. Such regions may be identified using segmentation, and a 3D printer could be used to print the malignant regions to replace the malignant bones.

Reviewer 2 Report

I like the idea of the paper and its presentation. It has covered most of the components of the machine learning model development process. 

I would like to see the comparison of your model with a similar paper in diagnostics. 

1. Efficient bone metastasis diagnosis in bone scintigraphy using a fast convolutional neural network architecture (https://www.mdpi.com/2075-4418/10/8/532)

Author Response

Point 1: I like the idea of the paper and its presentation. It has covered most of the components of the machine learning model development process. I would like to see the comparison of your model with a similar paper in diagnostics. 

Efficient bone metastasis diagnosis in bone scintigraphy using a fast convolutional neural network architecture (https://www.mdpi.com/2075-4418/10/8/532)

Response 1:  We thank the reviewer for their suggestion. We have added the suggested paper in the materials and methods section.

The authors of this paper propose an efficient approach that quickly classifies bone scintigraphy images of prostate cancer patients by determining whether or not they develop prostate cancer metastasis. The proposed method classifies the data into three categories: malignant, healthy, and degenerative. After various exploration and experiments, the proposed CNN architecture consists of 4 convolutional-pooling layers, 2 dense layers followed by a dropout layer, as well as a final output layer with three nodes. In the initial and the intermediate layers, ReLU activation is used followed by Adam optimizer at the output nodes. The results showed that the method is sufficiently accurate, with an accuracy of 91.61% when it comes to distinguishing a bone metastasis instance from other cases of degenerative alterations or normal tissue.

Round 2

Reviewer 1 Report

The authors have addressed the comments.